# Proof-of-Concept of Continuous Transfection for Adeno-Associated Virus Production in Microcarrier-Based Culture

Brian Ladd [1,2], Kevin Bowes [3], Mats Lundgren [4], Torbjörn Gräslund [2,5] and Veronique Chotteau [1,2,*]

1   Department of Industrial Biotechnology, School of Engineering Sciences in Chemistry, Biotechnology and Health, Royal Institute of Technology (KTH), 11428 Stockholm, Sweden; ladd@kth.se
2   AdBIOPRO, VINNOVA Competence Centre for Advanced Bioproduction by Continuous Processing, Royal Institute of Technology (KTH), 11428 Stockholm, Sweden; torbjorn@kth.se
3   Cobra Biologics, Charles Rivers, Keele ST5 5SP, UK; kevin.bowes@crl.com
4   Cytiva, 75323 Uppsala, Sweden; mats.lundgren@phase2phase.com
5   Department of Protein Science, School of Engineering Sciences in Chemistry, Biotechnology and Health, Royal Institute of Technology (KTH), 11428 Stockholm, Sweden
*   Correspondence: veronique.chotteau@biotech.kth.se

**Abstract:** Adeno-associated virus vectors (AAV) are reported to have a great potential for gene therapy, however, a major bottleneck for this kind of therapy is the limitation of production capacity. Higher specific AAV vector yield is often reported for adherent cell systems compared to cells in suspension, and a microcarrier-based culture is well established for the culture of anchored cells on a larger scale. The purpose of the present study was to explore how microcarrier cultures could provide a solution for the production of AAV vectors based on the triple plasmid transfection of HEK293T cells in a stirred tank bioreactor. In the present study, cells were grown and expanded in suspension, offering the ease of this type of operation, and were then anchored on microcarriers in order to proceed with transfection of the plasmids for transient AAV vector production. This process was developed in view of a bioreactor application in a 200 mL stirred-tank vessel where shear stress aspects were studied. Furthermore, amenability to a continuous process was studied. The present investigation provided a proof-of-concept of a continuous process based on microcarriers in a stirred-tank bioreactor.

**Keywords:** Adeno-associated virus; transfection; PEI; continuous; gene therapy; microcarriers; bioreactor; transient expression

## 1. Introduction

Gene therapy has the potential to be one of the next great revolutions in medicine. It allows for not only treatment but also potential cures for many debilitating diseases. Through the choice of vector, different tissues can be targeted making the treatment highly specific with few off-target effects. Interest in gene therapy has intensified with more than 1680 new drug trials being conducted from 2004 to 2017 [1].

One of the most promising vectors for the targeted delivery of therapeutic genes is the Adeno-associated virus (AAV) due to its highly specific targeting and lack of an immune response [2]. Most of the current AAV based therapies, are for rare diseases with low patient populations or have targets that require low doses. The current roadblock for the application of AAV based therapies for more prevalent diseases, or ones that require a high dose, is the limited manufacturing capacity [3]. An efficient scalable manufacturing platform for AAV vectors would allow for larger trials which would accelerate drug development and provide treatment options for diseases that affect larger populations.

Production methods of AAV based on transient transfection require plasmids encoding the viral proteins and DNA, which are required for vector assembly. Typically, the two open reading frames encoding the Rep proteins and the capsid proteins are placed on

the same plasmid (pRepCap) and the adenoviral genes necessary for AAV replication is placed on a second plasmid (pHelper); the gene of interest (GOI), flanked by the inverted terminal repeat sequences, is placed on a third (pGOI). However, there are designs that can reduce the number of plasmids [1–4]. The producer cells are grown either adherently or in suspension. Adherent cells are often able to provide a higher cell-specific production of AAV [2]. The popularity of the transient transfection method is due to the versatile production and rapid process development that it facilitates. The plasmids are transfected at the time of production and the specific serotype and GOI can be readily changed with only minor alterations to the process. This versatility allows for a quick transition between the production of different serotypes and GOIs.

The transfection agent that is most widely used is the cationic polymer, polyethylen-imine (PEI). The positively charged PEI facilitates transfection by forming a complex with the negatively charged plasmid DNA; this complex is called the polyplex. When added to the cells, the polyplex is taken up and, through endosomal escape due to the proton sponge effect of the PEI, the PEI-DNA complex is released within the cytosol where it can be transported into the nucleus. It has been shown that the size and properties of these complexes have a large influence on transfection efficiency [5–7], with smaller complexes being more readily taken up by the cell but having a lower chance of reaching into the nucleus and larger particles being taken up more slowly but are more effective in delivering their DNA payload.

Depending on the specific cell line and transfection conditions, a larger or smaller polyplex will lead to a more effective transfection. Therefore, it is necessary to reoptimize the transient transfection conditions when the mode of production changes or if a new cell line is used. In addition to the size of the polyplex, shear forces acting on the cells play an important role in the effectiveness of the transfection [8,9].

It has been shown that shear forces influence the uptake of nanoparticles including PEI-DNA complexes [8–10]. During transfection, the integrity of the cell membrane is compromised, increasing their susceptibility to damage by shear, meaning that conditions that previously did not affect cells could become damaging after transfection [9].

The cell-specific yields and vector quality are up to 15-fold higher in adherent systems compared to single-cell suspension systems [11]. While it has been shown that AAV vectors can be produced by cells in suspension with volumetric yields similar to processes with adherent cells, this is however, only achieved after a long and labor-intensive optimiza-tion [12,13]. Two-dimensional production systems, such as roller bottles or T-flasks, provide high specific productivity of AAV but suffer from scalability, making large scale produc-tion not economical [3]. The solution is to use a three-dimensional system that maintains the high cell-specific productivity of adherent cells but allows the process to scale with the volume rather than the surface area. If an adherent system could have the same cell density as a suspension platform, around $2 \times 10^6$ cells mL$^{-1}$, the volumetric yields would be significantly higher. Potentially, a microcarrier-based transient transfection process would combine the scalability of suspension cultures with the high specific productivity of adherent production, which could result in an extremely efficient process [14].

Microcarriers have been used since the 1960s for the cultivation of anchorage-dependent cells in a suspension system. Cultivations using microcarriers have been successfully used at scales in excess of 2000 L for the production of vaccines [15]. Microcarriers thus provide a possible solution to scale up the bottleneck of anchorage-dependent AAV production.

While being able to provide a close to suspension-like scalability, microcarriers have their own set of limitations. Among them is the increased sensitivity of the cells to shear due to the increase in effective size of the cell to the hydrodynamic environment [16]. Vortices, also called eddies, are formed in a turbulent fluid with the size of these eddies being proportional to the power transferred to the fluid. The smallest of these eddies and power input to the fluid are related by the Kolmogorov eddy length equation (1). When the eddy size decreases in a turbulent flow, the size between a particle and an eddy becomes comparable and the shear acting on the surface of the particle increases. If the Kolmogorov

eddy length is of comparable size to the microcarrier, high shear is experienced on the surface, damaging the attached cells [16].

The generation of a large volume culture with microcarriers is an issue for manufacturing. To enable an increase in the number of adherent cells, cells anchored on microcarriers need to be detached and reinoculated to a larger amount of microcarriers in a larger volume. This detaching and reattaching procedure presents a loss in efficiency compared to suspension cell culture.

Depending on the cell line, the anchorage dependency of the cells can be modified based on the culture medium used. If the anchorage dependency would be only required at a certain stage in the culture, during transfection and AAV production, for example, it could be more efficient and economically attractive to keep the cells in suspension during the previous stages, i.e., cell expansion, and then shift the culture to anchorage dependency on microcarriers at the production stage. Such a system lends itself very well to the production of AAV at scale because it can make use of the increased product quality and yield of adherent cells while maintaining the efficiency and economics of suspension culture to build the cell mass.

In the present study, it was hypothesized that HEK293T cells could be grown in suspension, offering the easiness of this type of operation and that the cells would then be anchored on microcarriers in order to proceed with the triple plasmid transient transfection in adherent cells for AAV expression. This latter process to benefit from a higher specific AAV yield, is often reported higher for adherent cell systems than cells in suspension. Bearing in mind that this process was developed in view of scaled-up commercial application in a bioreactor, the culture on microcarriers was studied in terms of limitations brought by shear forces in stirred tank vessels and the feasibility of continuous transfection. Finally, the present investigation aimed at providing a proof-of-concept of a process based on microcarriers in view of a readily scalable solution for the production of AAV at a commercial scale.

## 2. Materials and Methods

### 2.1. Culture Medium for Cell Expansion and Passaging

HEK293T cells (Cobra Biologics, Charles Rivers, ST55SP Keele, UK) were cultured as single cells in suspension and routinely passaged twice a week in Hyclone CDM4HEK293 medium, cat. No. SH30858.02 (Cytiva, 75323, Uppsala, Sweden). This medium was selected among four culture media for its ability to support the growth of cells in suspension. Using this medium, the specific growth rate, calculated using nonlinear least squares fitting of the total cell density, was $0.80 \text{ day}^{-1}$ against 0.61 to $0.70 \text{ day}^{-1}$ for the other tested media.

### 2.2. Medium for Transfection

A selection of the medium used for the triple transfection (presented in Section 2.3) among five media, A, ..., E, was carried out in static tissue culture T-25 flasks with a total end volume of 7 mL and a total DNA mass of 80 μg. The relative AAV titer was measured via CHO cell transduction (presented in Section 2.7). Supplementary Figure S1 shows the average transduction results normalized to the maximal observed expression obtained in medium D. From the transduction, the relative GFP expression indicated that the HEK293T cells transfected in medium D generated a greater production of the active vector than in the other media. Based on this result, it was decided that medium D would be used for all the transfection experiments. Medium D is a proprietary serum-free medium.

### 2.3. Adeno-Associated Virus Triple Transfection

The three plasmids for AAV production; pHelper, pRepCap, and pGOI (Cobra Biologics, Charles Rivers, ST55SP Keele, UK), were at a DNA concentration of $1 \text{ mg mL}^{-1}$ in TE buffer. The pGOI encoded green fluorescent protein (GFP) and the pRepCap were derived from AAV serotype 9. The transfection reagent was polyethyleneimine PEIpro (Polyplus), at a stock concentration of $1 \text{ mg mL}^{-1}$. The three plasmids, pHelper, pGOI encoding GFP,

and pRepCap, were mixed at a volume ratio of 2:1:1 with an end DNA concentration of 80 µg mL$^{-1}$ in the cell culture medium. The PEIPro was also diluted to a concentration of 80 µg mL$^{-1}$ in the cell culture medium. The DNA mix was then added to the PEIpro at a volume ratio of 1:1 (DNA: PEIpro ratio 1:1), briefly vortexed, and then left to incubate for 5 min before transfection in HEK293T cells. The number of passages before transfection did not exceed 30 passages. Prior to transfection, a 70% medium exchange was performed by stopping the agitation and allowing the microcarriers to sediment; after which, the appropriate amount of supernatant was removed and fresh medium was added. In the spinner flask transfections, the volume was reduced by 50%, from 50 mL to 25 mL. The volume was increased back to 100% 2 h post-transfection.

### 2.4. Cell Count and Viability Measurements

For all the transfection experiments and cell passaging/back-up maintenance, the density and viability of HEK293T and CHO cells were measured by BioProfile FLEX Analyzer (Nova Biomedical, Waltham, MA, USA) which uses the trypan blue exclusion method. For all the transduction experiments, the cell density and viability were measured by Norma XS (Iprasense, Clapiers, France), which is based on holographic imaging. All cell counts were made from samples taken directly from the culture without dilution.

### 2.5. Flow Cytometry

A Gallios flow cytometer (Beckman Coulter, Brea, CA, USA) was used for the quantification of the fraction of cells expressing GFP. The 488 nm excitation laser was used along with a 525 nm detector, channel 1, and a 575 nm detector, channel 2. An initial forward and side scatter gate was used to isolate the cell population. The fluorescence channels 1 and 2 were used to gate the cells expressing GFP by the constant ratio observed between these channels. All gating was done in the software Kaluza (Beckman Coulter, Brea, CA, USA); the data were exported to MATLAB (MathWorks, Natick, MA, USA) for further analysis.

### 2.6. Bioreactor and Spinner Flask Cultures

A DASBox system (Eppendorf, Hamburg, Germany) was used for the bioreactor experiment with 150 mL working volume, under feedback controls of pH, dissolved oxygen concentration, impeller speed, and temperature. The bioreactor was equipped with two marine impellers with a diameter of 3 cm [17]. The control software regulated the dissolved oxygen by varying the flow rate and proportion of air or pure oxygen at atmospheric pressure into the head space of the vessel. The pH was controlled by varying the flow rate of $CO_2$ into the head space (no upregulation of the pH was required). The agitation rate was controlled via an onboard tachometer. In the perfusion operation, a sedimentation tube with a diameter of 1 cm was used to retain the microcarriers.

Cultures in spinner flasks were performed using Bellco spinner flasks (Bellco Glass, Vineland, New Jersey USA), magnetically agitated at speeds specified in the text, in an incubator (37 °C, 5% pCO$_2$). The spinner flasks were equipped with a dual blade impeller, diameter 5.1 cm, height 2.27 cm.

The microcarriers Cytodex 3 (Cat. No. 17-5487-01) and Cytodex 1 (Cat. No. 17-5488-01) (Cytiva, 75323, Uppsala, Sweden) were prepared according to the manufacturer instructions. The spinner flask cultivations used gamma sterilized Cytodex 3 and for the bioreactor cultivations. Cytodex 1 or 3 microcarriers were prepared and then autoclaved for 20 min at 121 °C. Cytodex 1 and 3 microcarriers have a dry mass specific surface area of 4400 cm$^2$ g$^{-1}$ and 2700 cm$^2$ g$^{-1}$, respectively. A concentration of 3 g L$^{-1}$ was targeted, corresponding to a volume specific surface area of 13.2 cm$^2$ mL$^{-1}$ and 8.1 cm$^2$ mL$^{-1}$ for Cytodex 1 and 3. A target inoculation density of 50 cells per microcarrier, or $7.9 \times 10^4$ cells cm$^{-2}$ and $4.5 \times 10^4$ cells cm$^{-2}$ for Cytodex 1 and 3, was used unless otherwise specified.

### 2.7. Transduction Assay

For the determination of the amount of AAV produced, CHO cells were used for transduction. In a Corning 24 well deep well plate or tissue culture treated Corning 96 well plate, CHO cells were seeded to a final density of $0.1 \times 10^6$ cells $mL^{-1}$ in 1.5 mL or 150 µL of Ex-cell 302 medium from Sigma-Aldrich. 1 mL or 0.1 mL of AAV containing the sample was added to this culture and the plate was sealed with an adhesive filter and placed in a 37 °C, 5% $pCO_2$ incubator and shaken at 300 RPM with an orbital diameter of 2.5 cm or was statically incubated. After 48 h, the cells were analyzed by flow cytometry or by a fluorescence plate reader with an excitation wavelength of 500 nm, a dichroic filter of 520 nm, and an emission wavelength of 540 nm.

## 3. Results

### 3.1. Microcarrier Cultivation of HEK293T Cells

Medium D was selected among five media for its potential to support transfection and AAV production. This medium was also able to support the cell attachment and growth of microcarriers. A major benefit was that the medium used for growth on the microcarriers did not need to be changed to a different medium before transfection, which gave a great benefit to reduce the operations, a factor important for the scale-up of this process.

A preliminary study was dedicated to the characterization of the culture of the cells on microcarriers. Cells were inoculated in a Bellco 125 mL spinner flask with 3 mg $mL^{-1}$ Cytodex 3 microcarriers, at a cell density of $0.64 \times 10^6$ cells $mL^{-1}$ in a final culture volume of 50 mL at two different agitation rates. It was observed that a non-homogeneous distribution of cells on the microcarriers occurred for certain agitation rates. In Figure 1, the distribution of cells on the microcarriers is shown for two different agitation rates, the minimum speed allowing suspension, 50 RPM, and a higher rate, 70 RPM.

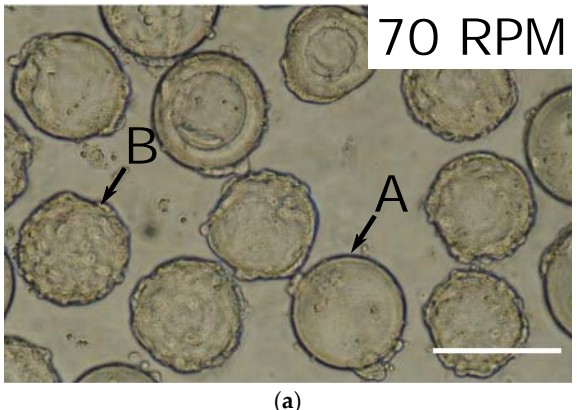
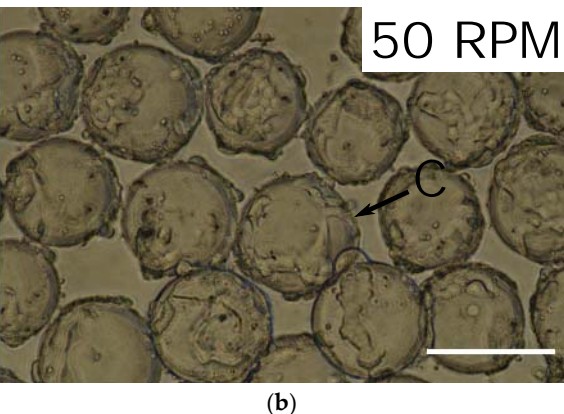

(**a**)　　　　　　　　　　　　　　　　　　　　　　　　　　　(**b**)

**Figure 1.** Pictures comparing the cell distribution on microcarriers of HEK293T cells inoculated at different agitation rates and a density of $0.64 \times 10^6$ cells $mL^{-1}$; the scale bar represents 200 µm. Arrows point at microcarrier bare from cells (A), fully confluent (B), and with homogenous cell distribution (**C**). (**a**) 70 RPM; (**b**) 50 RPM.

The cell distribution on the microcarriers was more homogenous when using 50 RPM agitation than 70 RPM. This was likely due to the fact that while a higher agitation rate led to greater mixing, it also brought more shear, which could detach cells from the microcarriers. Confluent microcarriers are better able to withstand this shear and are therefore preferentially sustaining cell growth compared to more sparsely populated carriers, accentuating the inhomogeneity. To minimize the effect of shear while maintaining the homogenization, the lowest agitation rate able to maintain adequate mixing should be used.

The agitation rate can greatly affect the attachment and growth of the cells on microcarriers [18]. The minimum rate able to homogenously suspend the microcarriers is known as the just suspended agitation rate or $N_{js}$ [19]. The minimum agitation rate to maintain

the microcarriers in suspension is affected by the reactor and impeller geometries as well as the medium properties and can be calculated with the Zwietering correlation [19].

$$N_{js} = S \times \nu^{0.1} \times d_p^{0.2} \times \left( \frac{g \times (\rho_s - \rho_l)}{\rho_l} \right)^{0.45} \times \frac{X^{0.13}}{D^{0.85}} \tag{1}$$

with $N_{js}$: Just suspended stirring speed (s$^{-1}$), $S$: Zwietering coefficient (–), $\nu$: kinematic viscosity (m$^2$ s$^{-1}$), $X$: solid loading fraction (kg solid kg$^{-1}$ fluid), $d_p$: microcarrier diameter (m), $D$: impeller diameter (m), $g$: gravitational constant (m s$^{-2}$), $\rho_s$: density of the microcarrier (kg m$^{-3}$), $\rho_l$: density of the fluid (kg m$^{-3}$).

In the present study, the minimum agitation rate for a 3 mg mL$^{-3}$ suspension of Cytodex 3 microcarriers was measured in both a 125 mL Bellco spinner flask and a DASBox reactor with dual marine impellers. The just suspended stirring speed in the spinner flask was measured to be 50 RPM and the Zwietering coefficient was calculated to 4.8, a value in line with the literature [20]. In the DASBox, the just suspended stirring speed was measured to be 150 RPM generating a Zwietering coefficient of 9.2. This larger Zwietering coefficient indicated that the DASBox system was less efficient in suspending microcarriers than the spinner flask for the present configurations of these vessels.

### 3.2. Hydrodynamic Comparison between Spinner Flasks and the DASBox Bioreactor

To determine the maximum shear limit acceptable for the cells on microcarriers for the bioreactor system, the agitation was slowly increased after the cells had attached to the microcarriers. This was applied from 150 RPM to 400 RPM by steps of 50 RPM in duplicate. Figure 2 shows photographs taken from both reactors, where the rows correspond to the reactor number, and columns a and b show samples taken from the 250 RPM and 300 RPM conditions. The attached cells were sheared off the microcarriers when the agitation rate increased from 250 RPM, column a, to 300 RPM, column b. This is most clearly seen by observing aggregated microcarriers; at 250 RPM cells were seen on the perimeter of the aggregates but at 300 RPM, column b, these perimeter cells had been sheared off.

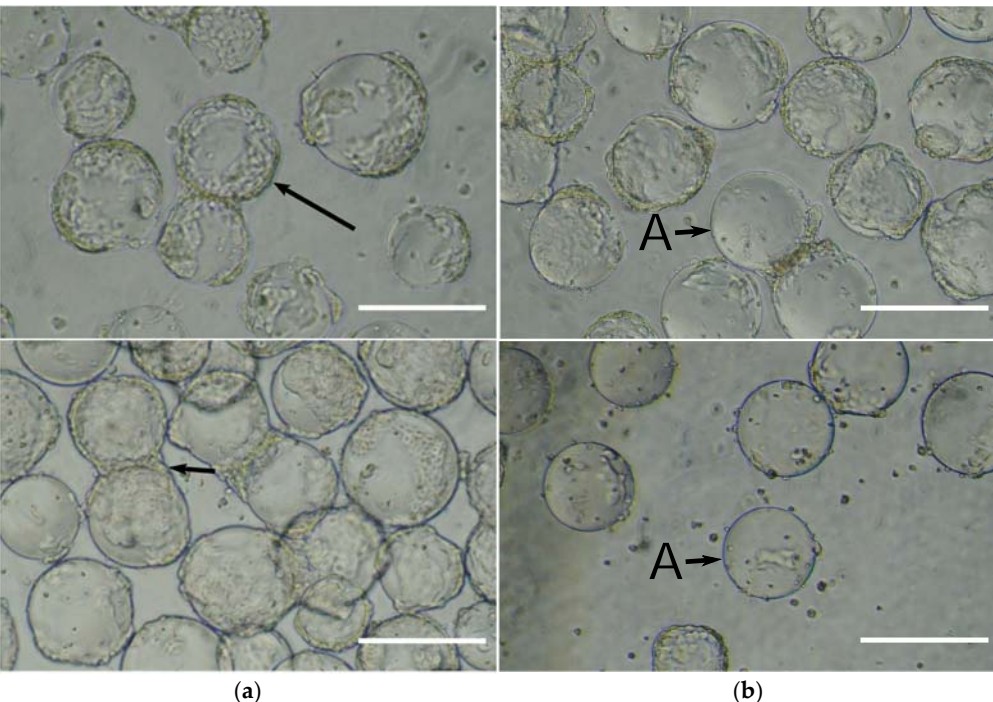

reactor 1

reactor 2

(a)　　　　　　　　　　　　　　　(b)

**Figure 2.** Determination of the maximum shear limit for HEK293T cells attached to Cytodex 3 microcarriers for the bioreactor system DASBox with dual marine impellers—experiment performed

in duplicate: reactor 1 (row 1) and reactor 2 (row 2); the scale bar represents 200 μm. The microcarriers were inoculated at 150 RPM; 24 h after inoculation, the impeller speed was slowly increased from 150 RPM to 400 RPM in steps of 50 RPM intervals. Images of the microcarriers are shown for ((**a**)—left column) 250 RPM where the cells were attached to the microcarriers, and ((**b**)—right column) 300 RPM where cell detachment from the microcarriers was observed. The unlabeled arrows represent cells well attached to microcarriers and the arrows labeled with A represent microcarriers where the cells have been sheared off by the hydrodynamic forces.

To make a detailed comparison between the spinner flasks and the DASBox; key hydrodynamic parameters were calculated for each vessel. The most important of these parameters is the Kolmogorov eddy length shown in Equation (2).

$$\sigma = \left( \varepsilon \times \left( \frac{\eta}{\rho} \right)^3 \right)^{\frac{1}{4}} \tag{2}$$

with $\sigma$: eddy size (m), $\varepsilon$: energy dissipation per mass (W kg$^{-1}$), $\eta$: dynamic viscosity of the fluid (Pa s), $\rho$: density of the fluid (kg m$^{-3}$).

Equation (2) requires the energy dissipation per mass, which can be calculated by Equation (3).

$$\varepsilon = \frac{Ne \times N^3 \times D^5}{V} \tag{3}$$

with $Ne$: dimensionless Newton number for the impeller (–), $N$: rotational frequency ( s$^{-1}$), $D$: impeller diameter (m), $V$: reactor volume (m$^3$).

The Newton number, $Ne$, was calculated for the spinner flask at agitation rates 50 RPM and 70 RPM using the Nagata correlation [21], see Table 1. The Newton number for the DASBox bioreactor was assumed to be identical to the value for a marine impeller as given in [22], also listed in Table 1.

**Table 1.** Newton numbers for the Bellco spinner flask and the DASBox bioreactor.

|  | Spinner Flask 50 RPM | Spinner Flask 70 RPM | DASBox |
|---|---|---|---|
| *Ne* | 0.52 | 0.46 | 0.36 [20] |

Using Equations (2) and (3) and the values in Table 1, the Kolmogorov eddy lengths were calculated for the spinner flask and the DASBox bioreactor and graphically represented as shown in Figure 3. Note that the power input for the DASBox was doubled to account for the dual marine impellers.

Croughan et al. determined that the area of cell damage begins when the eddy length decreases below 2/3 of the microcarrier diameter and an eddy length below 100 μm leads to rapid cell death [16]. These regions are marked in orange and red, respectively, in Figure 3. The DasBox was inoculated using an agitation rate of 150 RPM, allowing for even coverage and providing a low shear environment for them to develop a more robust attachment. In contrast, inoculation at 70 RPM in the spinner flask, an agitation point located within the cell damage region, leads to an inhomogeneous distribution of cells because of the increased shear. While the eddy length for 250 RPM in the DASBox bioreactor lies within the damage zone, the cells were able to withstand the shear. However, these conditions would most likely lead to negative effects if maintained for prolonged periods of time. The 300 RPM condition in the DASBox, another agitation point within the cell death region, leads to rapid cell loss. This is not surprising because at this agitation rate the eddy length is below 100 μm which is associated with cell death on microcarriers [16].

These values provided the limiting operating configurations for the spinner flasks and the bioreactor systems. Namely, inoculation should be at an agitation rate below the cell damage zone, 70 RPM for the spinner flask, and 233 RPM for the DASBox. Increases in

agitation up to 250 RPM in the DASBox can be tolerated, but prolonged exposure would most likely be detrimental. Finally, an agitation rate of 300 RPM and greater in the DASBox would shear cells off the microcarriers and result in cell death.

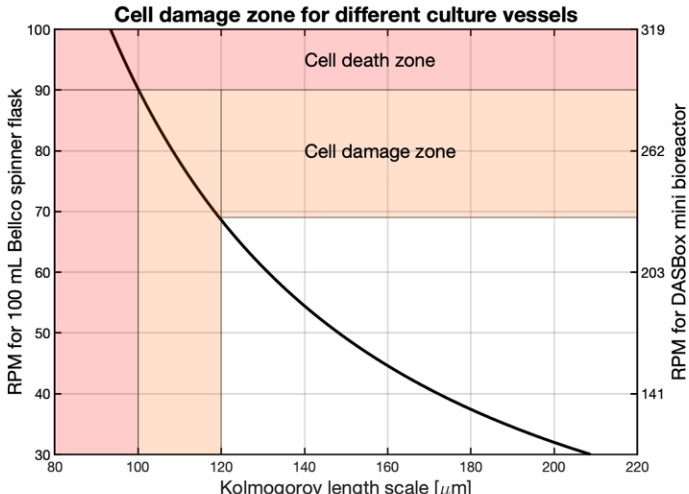

**Figure 3.** The Kolmogorov eddy length (x-axis) is represented as a function of the agitation speed in the Bellco spinner flask, left y-axis, and in the DASBox bioreactor, right y-axis, where the length scale associated with cell damage is marked in orange and the area associated with rapid cell death in red.

### 3.3. Triple Transfection of Cells Adherent on Microcarriers for AAV Production

#### 3.3.1. Mass Transfer Considerations for Microcarrier Based Transfections

Mixing during transfection improves the mass transfer between the bulk culture medium and the cells in comparison with static transfection. Thanks to the agitation, the PEI-DNA polyplexes only need to diffuse through a small film layer surrounding a particle instead of the liquid height to reach the cell surface. This causes a local increase in the concentration of PEI and DNA on the cell surface and could be a possible explanation of the improved nanoparticle uptake by cells under shear [10]. This local concentration can be calculated using the results from the shear studies performed in Section 3.

The Sherwood number represents the ratio of the convective mass transfer to the diffusive mass transfer and can be used to compare the relative effects of the different transport phenomena. Using the correlation in Equation (4), the Sherwood number, *Sh*, can be calculated for the case of microcarriers [23].

$$Sh = 2 + 0.4 \left( \varepsilon \times \frac{d_p^4}{\nu^3} \right)^{\frac{1}{4}} \times Sc^{\frac{1}{3}} \text{ where } Sc = \frac{\nu}{D_f} \tag{4}$$

$\varepsilon$: energy dissipation per mass (W kg$^{-1}$), $\nu$: kinematic viscosity of the fluid (m$^2$ s$^{-1}$), $d_p$: microcarrier diameter (m), $D_f$: diffusion coefficient of the PEI-DNA complex (m$^2$ s$^{-1}$), $Sc$: Schmidt number (–).

Using the Sherwood number from Equation (4), the normalized concentration of a chemical species on the surface of a microcarrier can be estimated for different cellular uptake rates [24] as follows;

$$X_N = \frac{X_b - X_c}{X_b} = \frac{\psi \times R_c \times d_p}{D_f \times X_b \times Sh} \tag{5}$$

$X_N$: normalized surface concentration (–), $X_b$: concentration in the bulk (mol L$^{-1}$), $X_c$: concentration on the surface (complexes m$^{-3}$), $\psi$: surface coverage of cells per unit area (cells m$^{-2}$), $R_c$: cell-specific reaction rate (complexes cell$^{-1}$ s$^{-1}$).

The concentration of PEI-DNA complexes was estimated by assuming a polyplex density similar to that of a protein, as studied in [25], and a polyplex radius of 300 nm, which was the average radius for a 5 min incubation reported in [26]. Equation (5) shows the normalized polyplex concentration as a function of the cell-specific complex uptake rate for a spinner flask operating at 50 RPM. Here, the cell-specific uptake rate was an estimate based on observations in several reports that the PEI-based transfection is complete after a maximum of 4 h, corresponding to an uptake rate of 0.01 complexes cell$^{-1}$ s$^{-1}$ [26–29].

The normalized concentration, see Equation (4), indicates the percentage decrease from the bulk concentration, where a normalized concentration of 5% means that the surface concentration is 95% of the bulk (100% − 5% = 95%). The normalized concentration gives an indication of where the majority of the mass transfer resistance lies. This is best illustrated by looking at the extreme case where the normalized concentration is either 1 or 0. When the normalized concentration is 1, this means that the surface concentration is 0, which can only be the case when the uptake rate of the cell greatly exceeds the transfer rate, i.e., the cell instantly takes up any polyplex that reaches the cell surface. Conversely, when the normalized concentration is 0, the concentration at the cell surface is equal to the bulk concentration, i.e., the transfer to the cell surface greatly exceeds the cells' ability to take up the polyplex Figure 4 shows that at the relevant uptake rates of between 0.01 and 0.05 complexes cell$^{-1}$ s$^{-1}$, the surface concentration (100%—normalized concentration) of the PEI-DNA complexes is between 75% to 95% of the bulk concentration. This means that the transport rate to the surface exceeds the uptake rate.

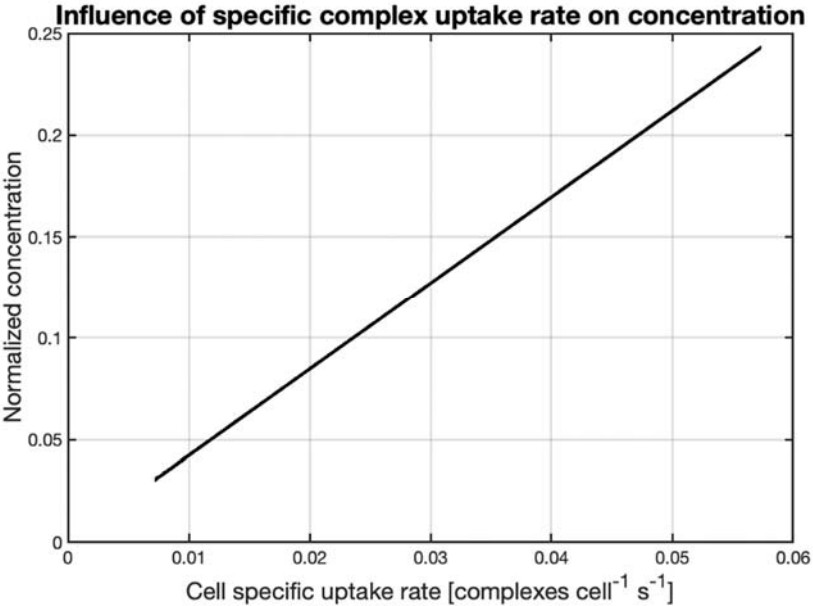

**Figure 4.** Theoretical normalized concentration of PEI-DNA complexes, calculated by Equation (5), as a function of the cell-specific polyplex uptake rate for a polyplex radius of 300 nm in a spinner flask stirred at 50 RPM. The cell-specific uptake rate was estimated based on PEI transfection completion obtained after a maximum of 4 h, corresponding to an uptake rate of 0.01 complexes cell$^{-1}$ s$^{-1}$. The PEI-DNA complexes concentration was estimated by assuming a polyplex density similar to that of a protein, and a polyplex radius of 300 nm, which was the average radius for a 5 min incubation.

The internalization mechanism mediated by heparin sulfate proteoglycans (HSPG) has been shown to affect the transfection efficiency of CHO cells, with lower numbers of HSPG binding sites resulting in lower transfection efficiency [30]. In addition, Mozley et al. showed that the rate of polyplex internalization was linked with the regeneration rate of HSPGs [30]. Therefore, if the rate of polyplex delivery to the cell exceeds the rate of this uptake mechanism, the polyplexes could either be internalized through another mechanism or be altered by the conditions in the culture; both will affect the transfection. It is then

important to consider not only the overall amount of DNA that is added but also the concentration of DNA at transfection.

### 3.3.2. Transfection in Spinner Flasks

HEK293T cells immobilized on Cytodex 3 microcarriers were transfected at two different DNA concentrations. Both DNA concentrations were chosen based on the preliminary transfection experiments in a static T-flask. The first condition was a final DNA concentration of 5.72 µg mL$^{-1}$ and the second concentration was taken identical to the static experiment, 11.44 µg mL$^{-1}$. The cell density at the time of transfection was $1.4 \times 10^6$ cells mL$^{-1}$ two hours post-transfection (2 hpt) the volume was increased from 25 mL to 50 mL, diluting the cell density down to $0.7 \times 10^6$ cells mL$^{-1}$. Both vessels were sampled daily, and the supernatant was used to transduce CHO cells to evaluate the production of biologically active AAV vectors. Figure 5a shows the fraction of transduced CHO cells as a function of the hours post-transfection (hpt).

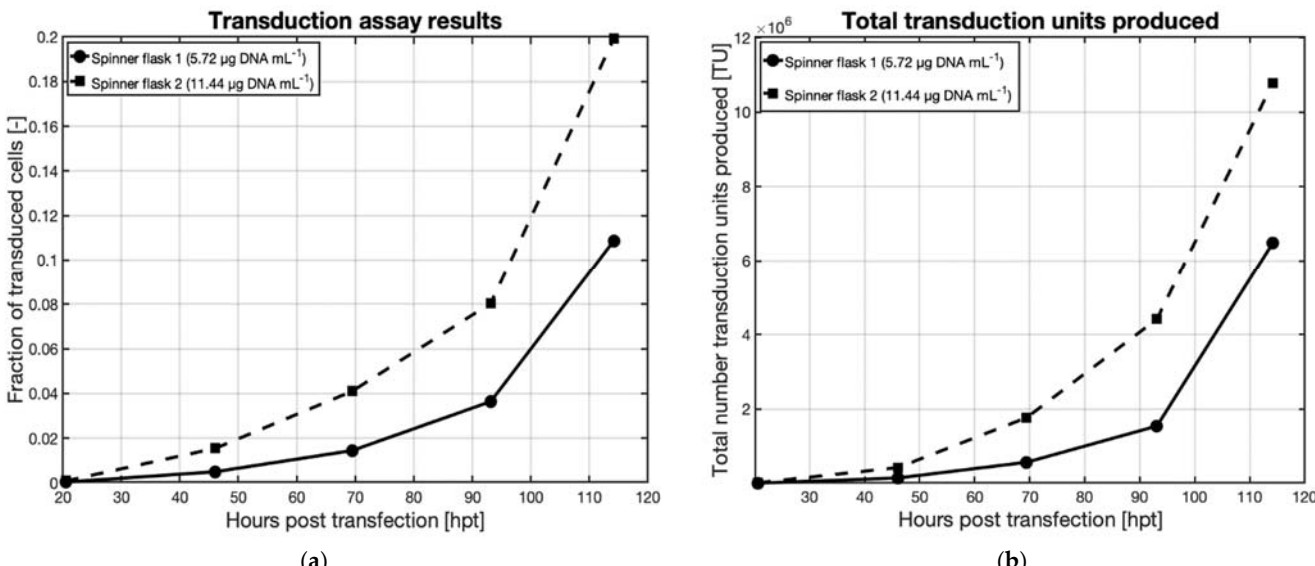

(**a**)        (**b**)

**Figure 5.** Effect of DNA concentration on the transfection of HEK293T cells immobilized on Cytodex 3 microcarriers in 125 mL Bellco spinner flasks, measured by transduction assays of CHO cells using supernatant samples harvested at various times from the spinner flasks with a cell density at transfection of $1.4 \times 10^6$ cells mL$^{-1}$; spinner flask 1 and 2 had final DNA concentrations of 5.72 µg DNA mL$^{-1}$ and 11.44 µg DNA mL$^{-1}$, respectively. (**a**) Fraction of transduced CHO cells from supernatant samples at selected points post-transfection and (**b**) Total number transduction units produced in both spinner flasks, calculated from Equation (6).

It can be seen in Figure 5a that twice as many CHO cells were transduced when using the supernatant from the highest DNA concentration, indicating that doubling the DNA concentration generated production of twice the amount of active AAV vectors. This suggests that at a concentration of 11.44 µg DNA mL$^{-1}$ the polyplex uptake mechanisms were not saturated. As a matter of fact, in case the uptake mechanisms had been saturated, the increase in the transduced fraction would have been smaller than the increase in DNA. It also indicates that the other mechanisms for AAV production were not saturated for this same reason. The latter condition, providing a superior outcome, was selected for further studies.

To quantify the number of active virus particles obtained at 114 h post-transfection the supernatants from both spinner flasks 1 and 2 were used to perform a Tissue Culture Infectious Dose 50 assay, TCID$_{50}$ assay, in which the supernatant samples were serially diluted and used to transduce CHO cells. Figure 6a,b represent, for transfected DNA concentrations of 5.72 µg mL$^{-1}$ and 11.44 µg mL$^{-1}$, the number of transduced CHO cells as

a function of supernatant volume used in the serial dilution. When volumes up to 0.25 mL of supernatant i.e., high dilutions, were used in the transduction assay, the amount of transduced cells increased linearly with the supernatant volume. In Figure 6a,b, lines fitted for this range of volume supernatant were drawn. The slopes of the values in the linear range of Figure 6a,b represent the change in the number of transduced CHO cells given the corresponding increase in the volume of supernatant added, in other words, it is a direct quantification of the concentration of active AAV in the supernatant. These slopes can be expressed in transduction units per milliliter [TU mL$^{-1}$], where one transduction unit is able to infect and cause one CHO cell to become GFP positive, and are $1.358 \times 10^5$ [TU mL$^{-1}$] and $2.143 \times 10^5$ [TU mL$^{-1}$] for spinner flasks 1 and 2 respectively.

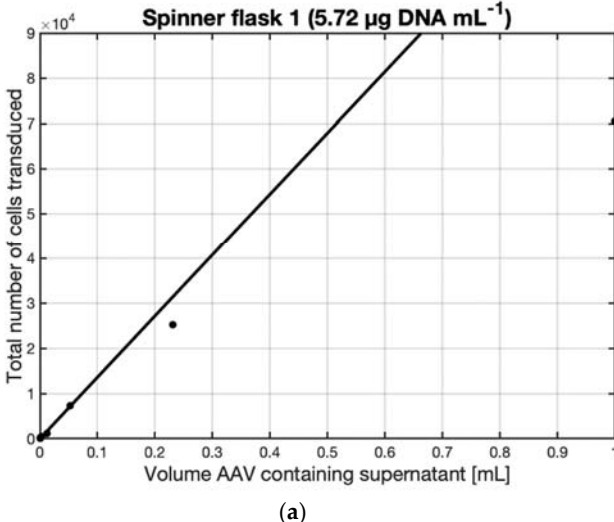 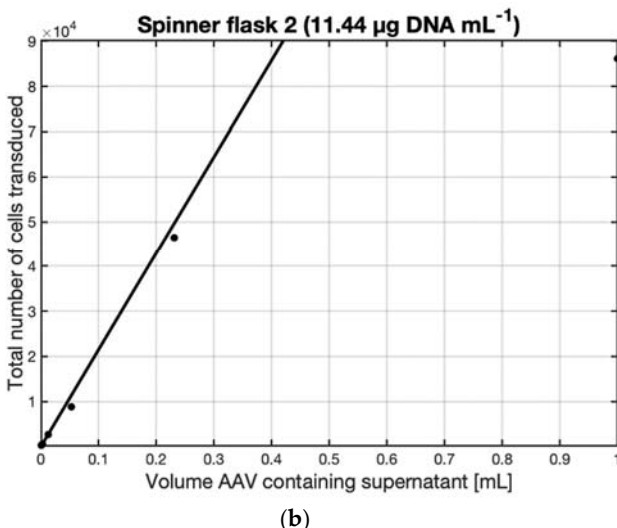

(**a**)  (**b**)

**Figure 6.** TCID$_{50}$ assay of supernatant samples harvested 114 h post-transfection from HEK293T cells immobilized on Cytodex 3 microcarriers transfected in 125 mL Bellco spinner flasks at a final concentration of (**a**) 5.72 µg DNA mL$^{-1}$ or (**b**) 11.44 µg DNA mL$^{-1}$ and a cell density of $1.4 \times 10^6$ cells mL$^{-1}$, corresponding to spinner flasks 1 or 2 of Figure 5a; supernatant samples were serially diluted and used to transduce CHO cells. The number of transduced cells increased with the supernatant volume until the number of cells used for the transduction was reached, leading to asymptotic behavior.

As can be seen in Figure 6a,b, the total amount of transduction units, measured by the TCID$_{50}$ assay, increased with the total number of transduced cells. This increase was linear for the lower values but approached an asymptote when the total number of transduced cells (y-axis) approaches the number of cells used for the assay. This behavior is expected as it would be impossible to transduce more than the number of cells in the assay. In Figure 7, the titers calculated in the TCID$_{50}$ assay were used to plot the total amount of transduced CHO cells vs. the total number of transduction units given in the transduction assay for both DNA conditions.

The data in Figure 7 fits a model with the function

$$y = \frac{b \times x}{(a - x)} \tag{6}$$

where $x$ is the total number of transduced cells and $y$ is the total number of transduction units with the parameters $a = 128{,}900$ and $b = 107{,}100$ determined by linear regression.

This relationship can be used to evaluate the number of transduction units from a CHO transduction assay culture, providing the titer of active AAV's. Figure 5b shows the total amount produced for both DNA conditions. The trend is very similar to the fraction of transduced cells, which is expected given that the same assay protocol was used for both assays.

This equation is specific for the transduction protocol used in this work and will most likely need to be recalibrated if parameters such as the medium containing the AAV, the cell number for the transduction, or the cell type are changed. It remains, however, extremely useful to monitor the titer of AAV during process changes.

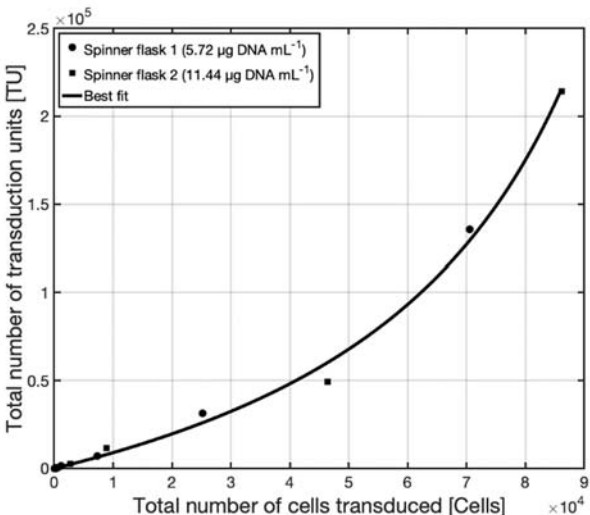

**Figure 7.** The total number of transduction units calculated from the titer determined by the $TCID_{50}$ assay, represented as a function of the total number of transduced CHO cells; approximation by model $y = b\, x\, (a - x)^{-1}$, coefficients $a = 128{,}900$ and $b = 107{,}100$ determined by linear regression.

### 3.3.3. Transfection in Bioreactor

The successful transfection of cells on microcarriers in spinner flasks showed that microcarrier cultures can be an effective tool to scale up AAV vector production for adherent cells. However, it was observed that shear could play a critical role in this process and therefore had to be taken into account in the up-scaling. To evaluate the importance of this factor on AAV production, a set-up mimicking a large-scale reactor was used. In the present study, a 150 mL DASbox bioreactor was operated with a perfusion system based on a sedimentation tube from which supernatant was pumped to the harvest tank while the microcarriers were retained in the bioreactor.

To limit the damage caused by shear, a rate of 175 RPM was used to closely mimic the conditions in the spinner flask at 50 RPM. While the DASBox operated slightly above the $N_{js}$, according to Figure 3, the Kolmogorov eddy length for the DASBox at 175 RPM was well below the cell damage zone and was similar to the shear environment in the spinner flask at 50 RPM. In these conditions, the Sherwood numbers (Equation (4)), or the ratio of convective mass transfer to diffusive mass transfer, were 35.71 for the DASBox (150 mL) and 33.00 for the spinner flask (50 mL). These values are similar, indicating that the mass transfer of the polyplexes to the surface of the cells was similar in these systems. In addition to the Sherwood numbers, the other variables of Equation (5) used to determine the normalized concentration gradient of the PEI-DNA complex were also similar in both systems. It was thus expected that the normalized surface concentrations of polyplex were similar between the spinner flask and DASBox.

Cells at a density of $1.5 \times 10^6$ cells $mL^{-1}$ were transfected in the DASbox bioreactor with a final DNA concentration of 11.44 µg $mL^{-1}$. Before transfection, the agitation was stopped, allowing the microcarriers to sediment; after which, 113 mL of supernatant was removed and 43 mL of polyplex mixture was added under agitation. Two hours post-transfection, 75 mL of fresh medium were added to restore the volume in the reactor to 150 mL. After transfection, samples were taken daily and the supernatant was used to evaluate the AAV production by transducing CHO cells, see Figure 8.

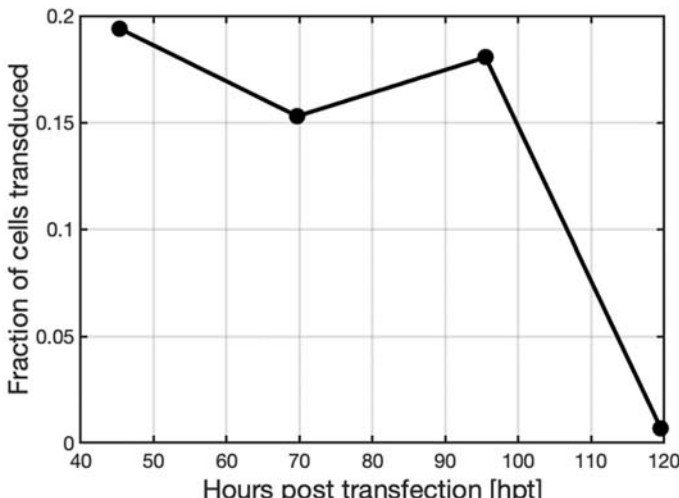

**Figure 8.** Transduction assay results of CHO cells using supernatant samples harvested at various time points from HEK293T cells immobilized on Cytodex 3 microcarriers after transfection at a final DNA concentration of 11.44 μg DNA mL$^{-1}$ in a 150 mL DASBox bioreactor operated in perfusion with a cell density at the time of transfection of $1.5 \times 10^6$ cells mL$^{-1}$; results indicated a stable AAV titer until day 4 followed by a decrease due to an increase of the perfusion rate.

It was observed that about 15% to 20% of the cells produced GFP when transduced with supernatant taken at days 2, 3, or 4 post-transfection, after which, the titer decreased steeply. The decrease in AAV titer on day 5 of the experiment was due to the perfusion rate being increased in the experiment and thus diluting the AAV.

Using the correlation of Equation (6) between the total number of CHO cells transduced and the total amount of transduction units, it is possible to calculate the accumulated volumetric productivity of the reactor, shown in Figure 9.

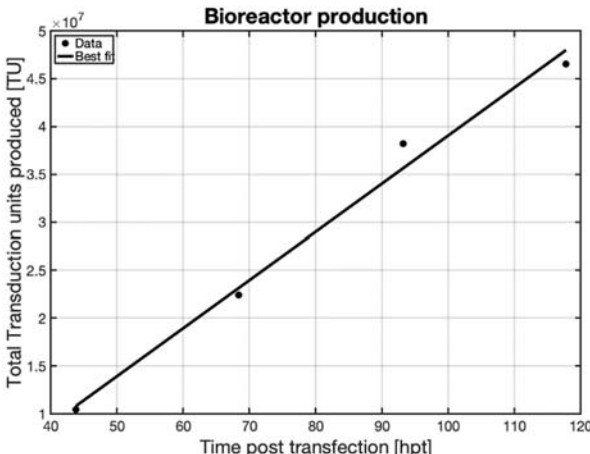

**Figure 9.** Total accumulated transduction units produced in the DASBox bioreactor operated in perfusion with a working volume of 150 mL and transfected with 11.44 μg DNA mL$^{-1}$, indicating that AAV was produced until the last day, day 5.

The total accumulated AAV production in the reactor was continuously increasing until day 5, after which the culture was terminated. The spinner flasks started at a low level of expression that increased exponentially over time, Figure 5b; whereas in the DASBox, the production increased linearly. The different production behaviors could be due to the differing environments in the DASBox compared to the spinner flasks, however, the productivity of the bioreactor was consistent with the spinner flasks. The total number of transduction units produced in the bioreactor was four times larger than the amount

produced in the spinner flask at the same DNA load, and used four times the volume of media, resulting in equal volumetric productivity.

### 3.4. Proof of Concept Continuous Transfection

On a large scale, the transient transfection process based on PEI requires transferring large volumes of mixed PEI and DNA which cannot be achieved in the same time frames that small- or micro-scale operations allow. The PEI and DNA mixture must incubate for a fixed amount of time to achieve the desired polyplex size. The time required to transfer large volumes of liquid will affect the delivered polyplex size; as such, this critical parameter will not be the same between the start and end of transfection in a batch reactor. Additionally, the use of a larger tank to prepare the polyplexes will take longer to homogenously mix and a larger distribution of polyplex sizes could occur because of the rapid condensation reaction. To counteract this, a continuous mode was adopted in the present work for the transfection, where a defined incubation time independent of the volume of the PEI and DNA was set.

A plug flow reactor is suited for a continuous transfection due to its narrow residence time distribution. To explore this concept, a continuous mixture of PEI and DNA followed by continuous transfection is proposed as follows. At the inlet of the reactor, the PEI and DNA are mixed, if the proper flow parameters are met, this can be achieved with a static mixer. This mixture then flows through the reactor while the PEI:DNA complexes form. At the outlet, the complexes exit after having incubated for a given time. This leads to a fixed complex formation time which is equal to the residence time of the reactor and is independent of the volume.

Furthermore, the continuous system needs to take into account the fact that the cells are adhering to microcarriers which requires two systems.

1. One system that can continually generate microcarriers with cells with a specific cell to bead ratio.
2. Another system to continuously transfect the cells on the inoculated microcarriers with the PEI:DNA complex.

System (1) was achieved with a combination of a static mixer and an intermediate stirred tank. Figure 10 shows a block flow diagram of this process.

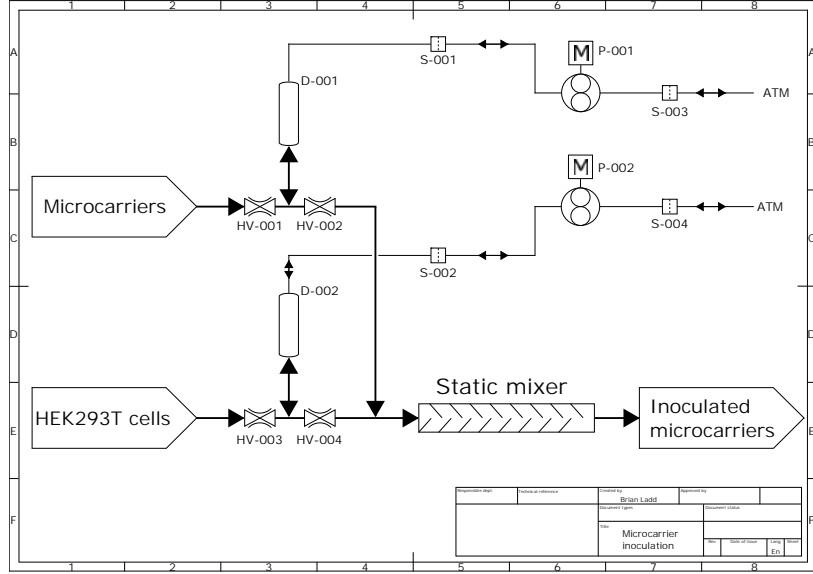

**Figure 10.** Process flow diagram of the continuous microcarrier inoculation system; D-001 and D-002 are 10 mL vessels; HV-001 to HV-004 are pinch valves; P-001 and P-002 are air pumps; S-001 to S-004 are 0.2 μm sterile air filters; ATM denotes atmosphere; DASBox bioreactor vessels are connected to labels "Microcarriers", "HEK293T cells" and "Inoculated microcarriers".

The microcarriers and cell suspension enter the diagram at the corresponding labels and were drawn by suction into vessels D-001 and D-002, respectively, and then the pinch valves HV-001 and HV-003 were closed. After the vessels D-001 and D-002 were filled, pinch valves HV-002 and HV-004 were opened and air pumped by pumps P-001 and P-002 was used to push the fluids through the static mixer into the intermediate reactor, denoted by the label "Inoculated microcarriers". The vessels D-001 and D-002 could hold up to 10 mL of liquid each and dispense it at a constant rate.

The static mixer shown in Figure 10 was built in-house with nine inserts made from curved stainless-steel inserts placed in a 5 mm ID silicon tube, similar in design to a Kenics type static mixer.

The static mixer was connected to three DASBox reactors, one contained sterile hydrated Cytodex 1 microcarriers at a concentration of 10 mg mL$^{-1}$ in medium (labeled "Microcarriers" in Figure 10), the second reactor had a suspension culture of HEK293T cells at $8 \times 10^6$ cells mL$^{-1}$ (labeled "HEK293T cells"), and the third reactor was the intermediate reactor connected to the outflow of the static mixer (labeled "Inoculated microcarriers"). A cell to microcarrier ratio of 100 was chosen which resulted in a microcarrier to cell volumetric flow ratio of 1:1. In two cycles, a total of 20 mL of both cells and microcarriers, for a total volume of 40 mL, was mixed at a total flow rate of 10 mL mL$^{-1}$. This flow rate was chosen to match the maximum power input for the DASBox using a marine impeller at 250 RPM, the highest agitation rate the cells were able to withstand on microcarriers. While this power input exceeded the limit for inoculation, the cells only experienced this stress for a brief amount of time. The maximum power input per unit mass for the DASBox was calculated using Equation (7) and the power input from a pipe flow was calculated using Equation (8), derived from the Darcy-Weisbach equation.

$$\varepsilon_{max,impeller} = \frac{20 \times Ne \times \rho \times N^3 \times D^5}{V} \tag{7}$$

$$\varepsilon_{tube\ flow} = \frac{512 \times \dot{V}^2 \times \eta}{\pi^2 \times d^6 \times \rho} \tag{8}$$

With $\varepsilon_{max,impeller}$: Maximum energy dissipation per mass for an impeller (W kg$^{-1}$), $\varepsilon_{tube\ flow}$: energy dissipation per mass for pipe flow (W kg$^{-1}$), $Ne$: Newton number, 0.36 for a marine impeller (–), $N$: rotational frequency ( s$^{-1}$), $D$: impeller diameter (m), $V$: Fluid volume (m$^3$), $\dot{V}$: fluid flow rate (m$^3$ s$^{-1}$), $\eta$: dynamic viscosity of the fluid (Pa s$^{-1}$), $d$: diameter of the tube (m), and $\rho$: Density of the fluid (kg m$^{-3}$).

Figure 11 shows a sample from the outlet of the static mixer directly after the first cycle of 10 mL microcarriers and cells.

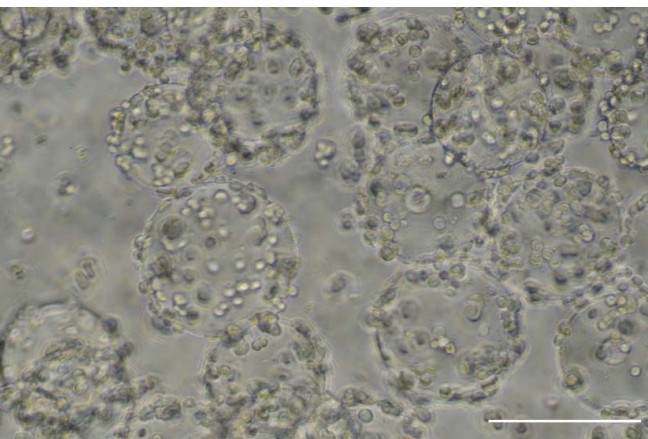

**Figure 11.** Image of the cell-loaded microcarriers taken at the exit of the static mixer of the continuous microcarrier inoculation process as depicted in Figure 10; the scale bar represents 200 μm.

It can be seen that the cells have attached well and demonstrate that a continuous stream of inoculated microcarriers can be obtained with a highly uniform distribution of cells on microcarriers.

At the small scales used here, a fully continuous transfection would require extremely low flow rates, on the order of µL min$^{-1}$, and would thus require a specialty microfluidic static mixer and very small diameter tubing. To alleviate this, a semi-continuous process can be implemented; Figure 12 shows a block flow diagram of the semi-continuous process that was developed.

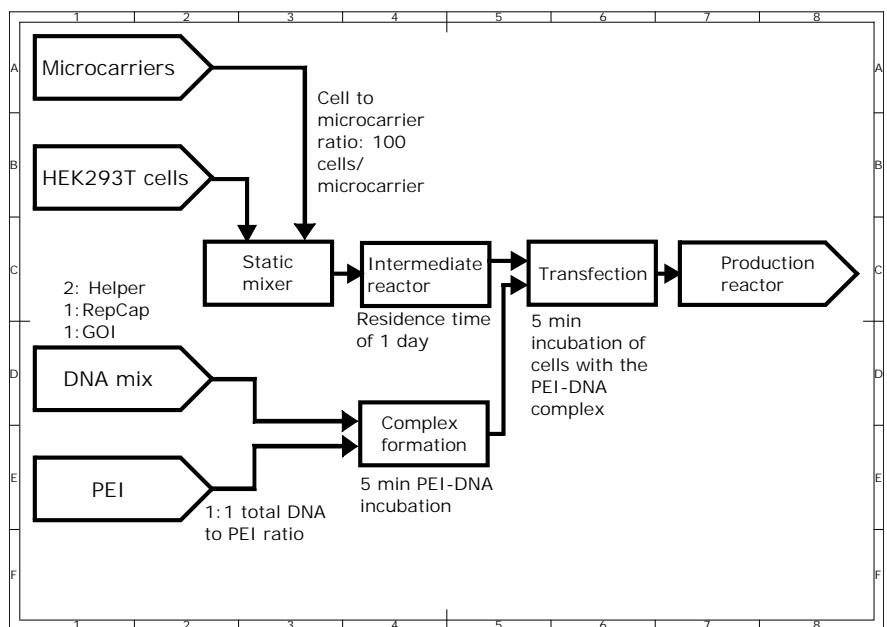

**Figure 12.** Block flow diagram of a semi-continuous transfection process. Microcarriers and HEK293T cells enter the process at the corresponding labels, where they are mixed in the static mixer; after which, the cells attach to the microcarriers. The inoculated microcarriers flow into the intermediate reactor which has a residence time of one day. The DNA mix and PEI enter the process at the corresponding labels where they are mixed at the given ratio in the complex formation step. The HEK293T cells attached to the microcarriers are then transfected with the formed complex, after which they are pumped into the production reactor.

In this process, the transfection reagents were mixed in a 100 mL Duran flask and allowed to incubate for 5 min; then, 50 mL of inoculated microcarriers were added and allowed to incubate for a further 5 min under gentle mixing. After this incubation, the transfected microcarriers were pumped into the production reactor using positive pressure from an air pump. The volume in the production reactor was kept below 200 mL. Prior to the addition of the transfected microcarriers, 50 mL was harvested from the reactor to maintain the volume. This process was repeated seven times over 20 days.

The long running time of this experiment would generate a large number of samples; to cope with this increased analytical demand, an adaptation was done to the transduction protocol presented above. The previous transduction protocol used suspension CHO cells and analyzed the GFP expression through flow cytometry; in the modified version, the same CHO cell type was used, but in a statically incubated 96 well plate measuring GFP expression through a plate reader. This modified protocol benefits over the previous transduction assay in two aspects; (i) the 96 well plate format is more conducive to the use of conventional multichannel pipettes, increasing the throughput; (ii) using a plate reader over a flow cytometer saves on sample preparation and running time, reducing the time needed to analyze many samples from hours to minutes. For these reasons, most of the samples of the semi-continuous transfection experiment were analyzed with the modified

transduction protocol, however, at selected time points the transduction protocol used in previous experiments was performed to harmonize the results.

The AAV production in the supernatant as measured by the previously used transduction protocol and the modified protocol can be seen in Figure 13a,b, respectively.

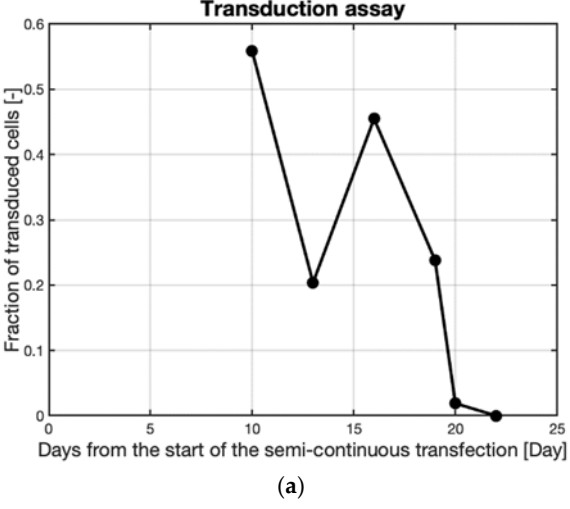

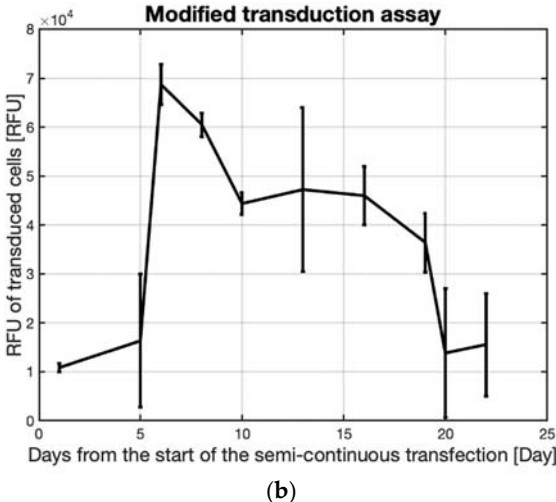

(**a**)

(**b**)

**Figure 13.** (**a**) Transduction assay results of CHO cells using supernatant samples harvested at various times during the continuous transfection, Figure 12, of HEK293T cells immobilized on Cytodex 1 microcarriers. (**b**) Modified transduction assay results of CHO cells using supernatant samples harvested at various times during the continuous transfection, Figure 12, of HEK293T cells immobilized on Cytodex 1 microcarriers.

The results of both transduction assays in Figure 13 show that the production of the active virus was maintained in the reactor for at least 16 days. When compared to the previous bioreactor transfection, the fraction of transduced cells is higher, indicating a higher titer. A drop in production was seen in the modified transduction assay after day 6 followed by a further decrease after day 16. This latter decrease could be due to a reduction of the transfection efficiency as the same mixing flask was used during the whole process, possibly subjecting the transfections to an accumulation of interfering components as some medium remained in the transfection flask after transferring the cells to the production reactor, influencing the composition of the complexation medium and leading to a less efficient transfection. The earlier decrease in production could be due to medium limitations in the production reactor. Fresh medium was added to the reactor after each transfection, but it is possible that in the altered state of virus production the 50 mL or 0.25 reactor volumes medium exchange was not sufficient. The presence of virus production until day 16 would suggest the latter decrease in AAV production was due to a change in transfection efficiency rather than medium limitation because medium limitation should have caused a decrease in production at an earlier time. This experiment shows that continuous transfection can be used to extend the production of AAV over a longer time than a traditional batch process.

## 4. Discussion

Gene therapy is set to revolutionize medicine. At the forefront of viral vectors, AAV is a leading candidate. The current roadblock for AAV is the production capacity needed for more extensive clinical trials and, later, supplying the commercial demand. A flexible production process is needed that can produce AAV vectors at different scales, depending on the demand. An ideal process would need to be easily scalable from vector screening all the way to large scale production. Adherent transfections, in addition to being a widely used AAV production strategy at the discovery phase, provide cell-specific titers up to 15 times that of suspension cells [11,31]. Current methods based on 2D cultures suffer from

the scalability needed to take an AAV therapy from discovery to commercial approval. Microcarriers provide the solution to this problem. A process using microcarriers can be readily implemented to existing transfection protocols without modification and thus does not need extensive optimization [14]. The work presented here shows that a 2D transfection process using T-flasks can be successfully scaled up to a 200 mL bioreactor with an effective surface area of over 1600 cm$^2$. A proof-of-concept continuous transfection is shown to be possible which will be able to increase the production capacity of a microcarrier-based system without increasing the footprint of the reactor.

The present system is a proof-of-concept showing that it is possible to continuously transfect cells adhering to microcarriers, themselves generated by continuous microcarrier inoculation. A fully automated transfection process could reduce the variability seen with manual transfections. Through a greater process control, the concentration of DNA, PEI, and perhaps even the size of the complex, could be dynamically changed to yield the most efficient transfection. A continuous transfection system could also be used to better understand the effect that certain medium components have on the production of AAV, e.g., by comparing the production rates in steady states of two different media compositions, allowing a direct comparison between both media. On the contrary, in batch mode, the dynamic nature can convolute the effects of certain components. In addition, and probably the most important issue for scale-up, at large-scale, a truly homogenous batch transfection is impossible since the homogenization cannot be instantaneous. The only way to generate a narrow distribution of PEI—DNA mixing time and therefore polyplex size, is to use a continuous transfection.

Looking at a scaled-up process of a 2000 L reactor scale performed in batch mode, the volume of PEI-DNA complex liquid needed to be transferred using the present transfection protocol is at least 500 L. To transfer the entire polyplex volume under a minute requires a pump flow of 30 m$^3$ h$^{-1}$ from a mixing tank of at least 500 L placed in close proximity to the culture bioreactor. Importantly, mixing in a 500 L size tank is potentially not evenly distributed, with areas of low mixing, e.g., near the impeller shaft, bioreactor wall, and bottom and liquid surface. DNA and PEI are large molecules, with a very low diffusion coefficient, which implies that mixing would mainly happen in the intense mixing zone, near the impeller. In batch mixing and operation, this localized mixing adds a great deal of heterogeneity into the amount of time the polyplexes have to react with each other and new PEI or DNA molecules, thus leading to a wide distribution in polyplex size.

As demonstrated here, the process can instead be carried out in continuous mode. When the involved liquid volume flow rates are larger than µL min$^{-1}$, the process can be transitioned to be fully continuous which provides easier operation and most likely more controlled transfection conditions for the cells. At a large scale, a homogenous single batch transfection leads to very large uncertainty, and the only way to have a narrow distribution of PEI:DNA complexation time is to use a continuous transfection. Furthermore, the continuous stream of inoculated microcarriers obtained here can also be scaled-up regardless of the amount needed, achieving a highly uniform distribution of cells on microcarriers.

**Supplementary Materials:** The following supporting information can be downloaded at: https://www.mdpi.com/article/10.3390/pr10030515/s1.

**Author Contributions:** Conceptualization, B.L., V.C. and T.G.; methodology, B.L.; formal analysis, B.L.; investigation, B.L.; resources, V.C., K.B. and M.L.; data curation, B.L.; writing—original draft preparation, B.L.; writing—review and editing, B.L., V.C. and T.G.; visualization, B.L.; supervision, V.C., K.B. and T.G.; project administration, V.C., K.B. and T.G.; funding acquisition, V.C. All authors have read and agreed to the published version of the manuscript.

**Funding:** This research was supported by the Competence Centre for Advanced BioProduction by Continuous Processing, AdBIOPRO, funded by Sweden's Innovation Agency VINNOVA (diaries nr. 2016-05181), by the *Marie Skłodowska-Curie European Industrial Doctorate programme EID, STACCATO*

*grant No 813453*, Cobra Biologics, Charles Rivers, for the cell line and the plasmid, and Cytiva for the culture medium.

**Institutional Review Board Statement:** Not applicable.

**Informed Consent Statement:** Not applicable.

**Data Availability Statement:** Not applicable.

**Conflicts of Interest:** The authors declare no conflict of interest.

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
