# Peer review of "Proof-of-Concept of Continuous Transfection for Adeno-Associated Virus Production in Microcarrier-Based Culture"

_processes, doi:10.3390/pr10030515_

Round 1
Reviewer 1 Report
Generally interesting study, some minor comments are listed below:
1) please include the catalog number for the materials used.
2) all figures are lacking descriptive captions to guild the readers through the results and summarize the figures each.
3) Font sizes needed to be enlarged for all figures
4) The formulation of the medium A-E is missing disabling the reader from reproducing the experiments.
5) Appearance changes of the cells in all the photos should be clearly annotated in the figure.
Reviewer 2 Report
Summary
Ladd et al. have developed a technique to improve vector production for recombinant Adeno-Associate Virus vectors. A major bottleneck in the efficient production and widespread of rAAV-based gene therapies is the ability to mass produce rAAV vectors. The authors report a proof-of-concept for a microcarrier-based production of rAAV vectors in spinner flasks which would enable vector production a commercial scale. In addition to producing virus in 3D spinner cultures, this method also contains a novel continuous transfection protocol. The authors optimized the hydrodynamic parameters, DNA concentration and conditions for continuous transfections in this study. This is a novel development in the field of gene therapy, and will facilitate mass production of vectors in the future. Pending some minor comments, the manuscript is suitable for publication in Processes.
Comments
- Line 51: Please rephrase “larger diseases populations” or correct the grammar. I am not sure what this means.
- Line 151: How was the AAV titer determined? Please be specific and provide primer sequences if qPCR was used for measuring titers.
- Figures 1, 2, 11: Need scale bars on the microscopy images.
